# Risk Exposure to *Legionella pneumophila* during Showering: The Difference between a Classical and a Water Saving Shower System

**DOI:** 10.3390/ijerph19063285

**Published:** 2022-03-10

**Authors:** Hélène Niculita-Hirzel, Audrey S. Vanhove, Lara Leclerc, Françoise Girardot, Jérémie Pourchez, Séverine Allegra

**Affiliations:** 1Department Work, Heath & Environment, Center for Primary Care and Public Health (Unisanté), University of Lausanne, Route de la Corniche 2, CH-1066 Epalinges-Lausanne, Switzerland; 2EVS-ISTHME UMR 5600, CNRS, University Jean Monnet of Saint-Etienne, F-42023 Saint-Etienne, France; audrey.vanhove@univ-st-etienne.fr (A.S.V.); francoise.girardot@univ-st-etienne.fr (F.G.); severine.allegra@univ-st-etienne.fr (S.A.); 3Mines Saint-Etienne, University of Lyon, University Jean Monnet of Saint-Etienne, INSERM, U 1059 Sainbiose, Centre CIS, F-42023 Saint-Etienne, France; leclerc@emse.fr (L.L.); pourchez@emse.fr (J.P.)

**Keywords:** *Legionella*, bioaerosols, shower systems, atomization technology, risk assessement

## Abstract

The increase in legionellosis incidence in the general population in recent years calls for a better characterization of the sources of infection, such as showering. Water-efficient shower systems that use water-atomizing technology have been shown to emit slightly more inhalable particles in the range of bacterial sizes than the traditional systems; however, the actual rate of bacterial emission remains poorly documented. The aim of this study was to assess the aerosolisation rate of the opportunistic water pathogen *Legionella pneumophila* during showering with one shower system representative of each technology. To achieve this objective, we performed controlled experiments inside a glove box and determined the emitted dose and viability of airborne *Legionella*. The bioaerosols were sampled with a Coriolis^®^ Delta air sampler and the total number of viable (cultivable and noncultivable) *Legionella* was determined by flow cytometry and culture. We found that the rate of viable and cultivable *Legionella* aerosolized from the water jet was similar between the two showerheads: the viable fraction represents 0.02% of the overall bacteria present in water, while the cultivable fraction corresponds to only 0.0005%. The two showerhead models emitted a similar ratio of airborne *Legionella* viable and cultivable per volume of water used. Therefore, the risk of exposure to *Legionella* is not expected to increase significantly with the new generation of water-efficient showerheads.

## 1. Introduction

Legionnaires’ disease (LD) is caused by the inhalation of water mists contaminated with *Legionella* spp., particularly *Legionella pneumophila* [1], and it represents a significant source of potentially preventable morbidity and mortality worldwide [2,3]. This disease often requires hospitalization of the patient and can be fatal for 5% to 10% of patients, even with antibiotic treatment [4,5,6]. In addition to causing this severe type of pneumonia, *Legionella* spp. are also known to induce a less severe flu-like condition known as Pontiac Fever (PF). The worldwide incidence of LD has continuously increased since 1996 [4,7,8,9], which calls into question the source of this phenomenon. Although *Legionella* bacteria are found naturally in freshwater environments, such as lakes and streams [10,11,12], they can become a human health concern when they grow and spread in human-made building water systems such as showerheads, sink faucets, cooling towers, hot tubs, decorative fountains and water features, hot water tanks, and heaters [13]. *Legionella pneumophila* colonization and persistence are mediated by biofilm formation, and survival within freshwater amoebae or *Caenorhabditis elegans* [14,15]. These bacteria are able to remain in the environment as free-living planktonic bacteria or form bacterial biofilms that adhere to surfaces [16]. *L. pneumophila* is able to enter the viable nonculturable (VBNC) state, which contributes to the resilience of this bacterial species under different harsh environmental settings [17,18]. This state can still promote host infection by *Legionella*, although the efficiency is lower than that of the mature intracellular form of *L. pneumophila* [19,20].

Certain criteria for certifying green buildings, such as lowering the hot-water temperature, as well as the unintended longer water residence time in the premise plumbing of these buildings, coupled with certain green plumbing devices available on the market in recent years, e.g., humidifiers, electronic faucets, low water consuming showerheads, have increased the risk factors associated with *Legionella* proliferation and emission in indoor air. A growing body of literature has focused on the risk of using such devices [21,22,23,24]; nevertheless the risk associated with *Legionella* emission by the last generation of water-efficient showerheads was not explored. Water-efficient showerheads are devices aimed reducing water and energy consumption. The water flow rate of showerheads drastically decreased from 9 L/min for classical showers to 6 L/min [25] and even 2.8 L/min for showers that use water atomizing technology [26]. To reach these ecological goals, water atomization showerheads generate water droplets smaller than that produced by classic shower systems [27], which can increase the risk of exposure to bioaerosols generated from water-associated contamination, such as the bacterium *Legionella*.

The inhalation of water aerosols containing *L. pneumophila* during routine showering has been regularly implicated in life-threatening respiratory infections, even before the advent of water atomization technology [28,29]. Up to 10% of particles generated during showering by classic shower systems have been previously shown to be small enough to deposit within the alveolar-interstitial region of the human respiratory system assuming oral and nasal breathing (6–10% and 0.9%, respectively) [30,31,32]. This number is strongly influenced by the shower system characteristics (number of nozzles, nozzle diameter, spray angle, water pressure, etc.), but also by the presence of a mannequin, which mimics the presence of the user under the shower [26,32]. We recently pointed out the differences between the two technologies in the behavior of inhalable water droplets emitted, both unobstructed and with the presence of a mannequin. While the mass of inhalable droplets increased with the presence of the mannequin during showering events with a continuous-flow showerhead [26,32], it decreased when a water-atomizing showerhead was used [26]. Moreover, in the presence of the mannequin, the inhaled water mass was reasonably consistent among the water atomization showers. However, the number of inhalable particles—in particular those that may carry bacteria—emitted by water-atomizing showerheads was generally slightly higher than that emitted by the classical ones.

To estimate the exposure risk associated with the inhalation of bioaerosols generated by these new types of showerheads, a quantitative microbial risk assessment (QMRA) framework is required from exposure to risk characterization, including biofilm impacts, aerosol generation, bacteria survival and transport within size-resolved water droplets, and interaction with a human receptor [33]. Therefore, in addition to data on aerosol generation, data on bacteria survival and transport within size-resolved water droplets during showering are needed. While the number of inhalable water droplets emitted in the range of bacterial sizes has been estimated, as well as the proportion of those carrying an infectious bacterium and able to reach the pulmonary alveoli (0.7% of the initial viable bacterial suspension) [34], there is still a need to assess the survival and transport of *Legionella* by realistic shower systems under controlled conditions.

Our aim was to experimentally determine, using a controlled system, the number of viable or cultivable *L. pneumophila* emitted during showering with one continuous flow and one water atomization showerhead previously shown to emit a distinct number of water droplets. The secondary aim was to estimate the impact of the water atomizing technology on *Legionella* survival. Therefore, we used the previously developed experimental systems to perform a risk assessment of infectious *L. pneumophila* [34,35,36].

## 2. Materials and Methods

### 2.1. Design of the Experiment

Water with a known concentration of *L. pneumophila* was pumped from a reservoir through a shower system with continuous flow or water atomizing technology inside a glove box to confine *L. pneumophila* dispersion during experiments. First, to test for *L. pneumophila* survival in water, we sampled equal volumes of water exiting the nozzle of each shower unit. Second, to determine the proportion of aerosolized bacteria and their physiological state in the aerosols, we sampled the bioaerosols generated during a short showering event by each showerhead with a Coriolis^®^ Delta air sampler (Bertin Instruments, Instruments, Montigny-le-Bretonneux, France). The total number of bacteria emitted and the bacterial viability were determined in all samples by flow cytometry, although the proportion of cultivable bacteria was determined by culturing on BCYE agar plates supplemented with chloramphenicol. Three showerheads of each technology were systematically used.

### 2.2. Description of the Experimental Setup

We selected two commercial showerheads, one continuous flow and one using the water atomization technology, among previously tested showerheads that emit the most water droplets in the bacterium size range [26]. The characteristics of these two showerheads models, namely STA for the continuous flow water technology and ECO for the water-atomizing technology, are described in Table 1.

The showers were run inside a 815-PGB glove box “La petite glove box” at atmospheric pressure (variation between glove box and exterior environment did not exceed 0.5 kPa (0.077 PSI)). A Fisher Scientific pump model STERWINS-900 W, 3800 L·h^−1^, 4.3 bars (Leroy Merlin, France) was used to draw the *Legionella* suspension from a 4 L reservoir through a flexible hose (150 cm) and showerhead (STA or ECO) at a water flow similar to that observed in the real sized shower cabin used for showerheads selection [26]. The water coming out from the showerhead (~2.75 L) was collected in a bag. For the experiment testing *Legionella* survival in water, the bag was completely closed around the showerhead (Figure 1a), and for the experiment that controlled for bioaerosols emission, the bag was open (Figure 1b). To collect equal volumes of water in the limits of the capacity of the reservoir in the glove box, the duration of the showering with the STA was 15 s, and that with the ECO was 30 s. The bioaerosols were collected in 3 mL of sterile distilled water at 10 ± 2 cm from the showerhead with the Coriolis^®^ Delta high air volume collection tool (Bertin Instruments, Montigny-le-Bretonneux, France). Thus, the Coriolis tool was at a distance where the emitted aerosols were previously proven to be homogeneous in the air [26]. Moreover, the Coriolis tool has been shown to not be sensitive to fluctuations in water droplet size induced by their evaporation between 10 and 60 cm. The sampler flow rate was 150 L/min. To collect aerosols from the overall volume of the glove box, the air sampling duration was extend by two additional minutes compared to the duration of the shower event (2 min 15 s or 2 min 30 s) depending on the showerhead used. The experiment was run in triplicate with three distinct showerheads STA or ECO.

### 2.3. Strain

Although there are over 50 species of *Legionella* [2] and a growing list of at least 16 serogroups of *L. pneumophila* [37], the majority of human infections are caused by *L. pneumophila* serogroup 1 [38]. Consequently, we chose to experimentally use a *L. pneumophila* serogroup 1 strain expressing a green fluorescent protein the Lp1 008-GFP strain, for which we previously optimized the culture and flow cytometry analysis protocols. This Lp1-008 GFP strain has the same behaviour as the original strain isolated from a hospital water network. A calibrated suspension (CS) of Lp1 008-GFP was prepared in sterile filtered tap water, and the quality is described here. Lp1 008-GFP was cultured on BCYE agar (Buffered Charcoal Yeast Extract, SR0110A, Oxoid, Dardilly, France) plates supplemented with 8 µg·mL^−1^ of chloramphenicol (Sigma C0378) at 36 ± 2 °C. Three day-old colonies were scraped from BCYE plates and resuspended in 50 mL of sterile distilled water to obtained a mother solution with an optical density of 0.45 at 600 nm (Colorimeter, Colour wave CO 7500, VWR France 633-0160). Then, the suspension was diluted into a bucket (reservoir) containing 4 L of sterile filtered tap water to obtain a working concentration between (4.5 ± 1.0) × 10^4^ and (5.5 ± 1.0) × 10^5^ CFU·mL^−1^. After performing sampling for culture and flow cytometry, the remaining output water was decontaminated by filtration twice though a 0.45 µm membrane to avoid environmental contamination with Lp1 008-GFP.

### 2.4. Sterile Filtered Tap Water

The water used for the calibrated suspension was tap water from the Centre for Health Engineering of Mines Saint-Etienne (CIS, Saint-Etienne, France) which was filtered through a 0.45 µm cellulose nitrate filter (Sartorius 11406-47-ACN) prior to each experiment. Before adding the bacteria, several parameters were checked. The total and free chlorine were quantified using the USEPA DPD (*N*,*N*-diethylparaphenylenediamine) method 8167 and 8021, respectively, as recommended by the manufacturer (Hach Company). Briefly, to determine the concentration of total chlorine, the content of one DPD Total Chlorine Powder Pillow (Hach 21056-69) was added to 10 mL of filtered water and, after 3 min, the absorbance was measured at 520 nm with a colourimeter (Hach DR890). To quantify the concentration of free chlorine, the content of one DPD free chlorine powder pillow (Hach 21055-69) was mixed with 10 mL of water and placed immediately into the cell holder to measure the absorbance. The total and free chlorine concentrations observed were 0.02 and 0.03 mg·L^−1^, respectively. To test the hardness of the filtered water, a hardness dip strip (Aquadur 912 20) was used. The concentration of CaCO_3_ was 0.7–1.2 mol·m^−3^, indicating soft water. The temperature of the water in the reservoir was 38 ± 2 °C at the beginning of the experiment.

### 2.5. Cultivability

Before starting the pump, 1 mL of the reservoir was collected and immediately serially diluted (1:10) in sterile distilled water. Two hundred microliters of dilutions 10^−2^ and 10^−3^ were plated in duplicate on BCYE agar supplemented with chloramphenicol. At the end of the experiment, the same method was applied to 1 mL collected from the bag. For the quantification of *Legionella* in the bioaerosols, 200 µL of the Coriolis collection tube was plated in duplicate onto BCYE medium supplemented with chloramphenicol. After 3 days of incubation at 36 ± 2 °C, the colonies were enumerated with a Scan 4000 instrument (Interscience, Puycapel, France).

### 2.6. Flow Cytometry Assay

The total number and viability of *Legionella* were determined in the reservoir, bag, and bioaerosols by flow cytometry assays (FCAs) at 3 ± 1 h after sampling. As previously described [35,39], 1 mL of each sample was directly labelled with 5 μL of propidium iodide (PI) at 1 mg/mL (Invitrogen P3566) in the dark for 5 min. The FCA profiles of the samples (Figure 2) were obtained using a combination of GFP green fluorescence (viable cells expressing GFP, i.e., viable and cultivable (VC) and viable but noncultivable (VBNC)) and propidium iodide (PI) red fluorescence for cells with damaged membranes (VBNC and dead cells (DC)). Flow cytometric measurements were performed using a CyFlow Cube6 instrument (Sysmex Partec, Görlitz, Germany) equipped with an aircooled argon laser (488 nm emission; 20 mW). The green fluorescence emission from GFP was collected in the FL1 channel (500–560 nm), and the red fluorescence from PI was collected in the FL3 channel (670 nm). A threshold was applied to the FL1 channel to eliminate background signals. Analyses were performed at a low-flow-rate setting at 40 µL. The results were analysed with Flowing Software 2.5.1 (Turku Bioscience, Turku, Finland).

### 2.7. Statistical Analysis

The data collected from the water samples, which are expressed in total cell counts (TCC)·mL^−1^ or colony forming units (CFUs)·mL^−1^, were directly used for the statistical analysis.

The data collected from aerosol samples, given by the flow cytometer in TCC·mL^−1^ or CFUs·mL^−1^, were transformed in TTC·m^−3^ or CFUs·m^−3^ according to the formula (1) where *Ca* is the number of counts or CFUs of *L. pneumophila* detected per mL of sample, *Vl* corresponds to the 3 mL in which the bioaerosols were sampled and the *Va* corresponds to 0.3375 m^3^ or 0.375 m^3^ aerosol volume collected for STA and ECO, respectively.
(1)$$\frac{\;Ca\times Vl\;}{\;Va}\;$$

The proportion of *L. pneumophila* emitted from water in the air during showering was determined using formula (2).
(2)$$\frac{\;Cw\times Vw}{\;Ca\times Vl}$$
where *Cw* is the number of counts or CFUs of *L. pneumophila* detected per mL in the water, *Vw* is the volume of water used during the shower event, 255 mL or 275 mL for STA and ECO, respectively, *Ca* is the number of counts or CFUs detected per mL of aerosol sample and *Vl* corresponds to the 3 mL in which the bioaerosols were sampled.

Descriptive statistics are presented as means with standard deviations (SD) and were used to illustrate the number of bacteria in samples. Paired *t* tests were conducted to determine whether there was a statistically significant difference between the samples collected before and after showering. Kruskal–Wallis nonparametric tests were used to identify differences in bacterial emissions between showerheads with different technologies. A *p* value of less than 0.05 was considered statistically significant. All analyses and graphs were carried out using STATA 14 software (StataCorp LLC., College Station, TX, USA).

## 3. Results

### 3.1. Validation of the Experimental Setup

The water pressure of the experimental setup was normalized for each shower system to the values observed previously in the real-sized shower cabin by choosing an adequate pump. The impact of tap water quality on *Legionella* survival in the reservoir was determined by flow cytometry, and no significant effect was observed (data not shown). The efficiency of collecting bacteria after showering in the bag was supported by the similarity in the concentration of total bacteria observed in the water samples (TTC·mL^−1^) collected before and after showering, (1.8 ± 0.7) × 10^6^ and (1.9 ± 0.6) × 10^6^ bacteria·mL^−1^, respectively (Figure 3).

### 3.2. Impact of Water Technology on Legionella Viability in Water

The impact of water technology on *Legionella* viability was determined by comparing the proportion of viable bacteria after passing through the shower system using water flow technology (STA) to that using water atomization technology (ECO). Most bacteria were resistant to the flow and pressure of both showerhead models (Table 2). No significant difference in the total cell numbers, viable fraction, or cultivable fraction was observed between the two shower systems (Table 2, Figure 3 and Figure 4).

### 3.3. Impact of Water Technology on Legionella Aerosolisation

The total number of airborne bacteria emitted during showering was determined for an equivalent volume of water used for STA and ECO (Figure 3). While 73% to 74% of the aerosolized bacteria were viable (VC + VBNC) only 2% to 3% were cultivable (Table 2, Figure 4). The cultivability rate was also low in the water samples (Table 2). For an equal volume of water, no significant difference in the number of bacteria emitted, regardless of their physiological state, was observed between the STA and the ECO showerheads. In addition, the ECO showerhead did not significantly decrease the bacterial survival and cultivability in the aerosols compared to the STA showerhead (Table 2). The proportion of total bacteria aerosolized from the total bacteria present in the overall volume of water was determined to be 0.02% ± 0.003%. The proportion of viable bacteria emitted during showering reached 0.02% ± 0.009% of the overall viable bacteria running through the shower system. Nevertheless, the proportion of cultivable bacteria aerosolized was estimated to be as low as 0.004% ± 0.0007% from the overall cultivable bacteria content in water samples, and only 0.0005% ± 0.00008% from the overall bacteria detected in water samples.

## 4. Discussion

In the present study, we experimentally compared the number of *L. pneumophila* emitted during showering via the dispersal of water droplets by a showerhead with continuous flow technology with that emitted by a showerhead with the water atomization technology. We showed that both showerheads emitted similar levels of bioaerosols that represent only 0.02% of bacteria present in water. Although 0.02% were viable, only 0.0005% were cultivable. This proportion is in the same range as that previously described in other house shower systems [40,41] and cooling systems [42].

One of the main findings of our study is that the majority of aerosolized bacteria are in a VBNC state, while a small proportion of the inhalable water droplets emitted during a shower event carry a highly infectious bacterium. A growing body of evidence indicates that the cultivable fraction of *L. pneumophila* does not represent all infectious propagules. The VBNC state of *L. pneumophila* induced by environmental stress [39,43,44,45] appeared to remain active and maintain low metabolic activity, membrane integrity, virulence-related protein expression [44,46], and low virulence levels [20,47,48], even if the bacterium could not grow on standard media. Therefore, it is important to estimate the viable fraction emitted to assess the risk of exposure, and not to limit the estimate to the cultivable fraction, when modeling of Legionella exposure is considered.

Another major finding is that the number of bacteria emitted did not differ between the two shower systems in spite of the difference observed in the number of water droplets emitted [26]. This can be explained by the small number of water droplets that can really carry a bacterium.

However, our results did not support a higher impact on survival or physiological state of *Legionella* by atomization technology compared with classical technology, as hypothesized. One reason might be that *Legionella* is in a planktonic state. It will be interesting to estimate the effect of the high-speed impact of water on the surface of the shower nozzle which is specific to atomization technology, on the physiological state and release of *Legionella* from the natural biofilm formed in hoses.

In the present study, we have chosen to estimate the bioaerosol emissions from a homogeneous solution of *Legionella* of known concentration to be able to compare shower emission between distinct showerhead systems. Previously published studies were based on the natural concentration of *Legionella* in water which did not allow for the control of the concentration throughout the shower event. Another novelty of our study was to assess the risk of exposure under the worse scenario, namely, exposure to water that is highly contaminated with *Legionella*. While such concentrations of *Legionella* were rarely reported in shower systems, they were common in cooling towers. However, some limits need to be mentioned. The infection risk linked to *Legionella* and consequences on human health and the environment required the use of only a limited volume of water. This fact has a direct consequence on the showering duration, which was much shorter than that of regular shower events (e.g., 10 min). Under such time limits, only 3 L of water passed through each shower system, which affected the bioaerosols emitted. Moreover, it is generally hypothesized that the bacteria released from biofilms are not homogeneous during a showering event. The release is expected to be higher in the initial volume of water compared with that after 5 min. Nevertheless, previous studies have not estimated the length of time to run the shower or the water volume that is required to decrease the concentration of bacteria to a level that is not detectable in aerosols. Another constraint imposed by the *Legionella* biohazard was the distance from the showerhead for bioaerosol sampling. This distance, which was approximately 10 cm, is much shorter than the distance between the user nose and the showerhead (generally approximately 40 to 60 cm). The sampling distance may affect the number of water droplets that can evaporate before reaching an inhalable size and thus biases the risk assessment performed in the present study. However, such an effect might be quite limited, as previously suggested by the homogeneity in the number of inhalable water droplets between 10 and 60 cm for the shower [26]. Finally, the biohazard linked to *Legionella* handling also constrained us to work with a bacterial solution despite growing the bacteria on a biofilm. Indeed, in the present study, we could not estimate the effect of water atomization technology on *Legionella* release from biofilms and compare it to that induced by a regular water flow shower system. Nevertheless, it will be highly interesting to explore such an impact on naturally developing biofilms with or without *Legionella*.

## 5. Conclusions

During the first flash of a showering event, up to 0.02% of *Legionella* can be aerosolized, and although most of the bacteria are in a VBNC state, only 0.0005% cultivable. The tested technology had no effect on *Legionella* viability. Thus, for a constant volume of water, a similar number of physiologically active bacteria are emitted by the two showerheads tested. Therefore, using ecofriendly showerheads should not increase the risk of being infected by *Legionella* during showering. Nevertheless, further studies are needed to assess whether these two technologies have a similar impact on the growth of biofilms in hoses and whether they induce the release of viable bacteria at similar rates.

## Figures and Tables

**Figure 1 ijerph-19-03285-f001:**
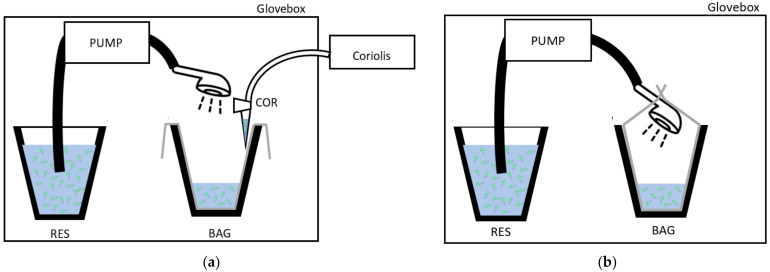
Experimental setup inside the glove box design (**a**) to collect the water contaminated with *Legionella* in controlled conditions in a bag (BAG); (**b**) to collect the bioaerosols during showering with a Coriolis^®^ Delta air sampler in a water sample (COR). RES: *Legionella* suspension reservoir.

**Figure 2 ijerph-19-03285-f002:**
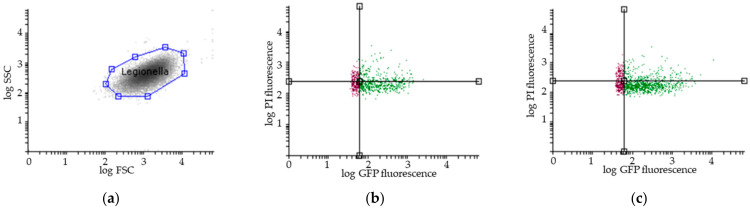
Representative results of flow cytometry assay (FCA). (**a**) *Legionella pneumophila* population gating, SSC (Side Scatter), FSC (Forward Scatter); Viability FCA profiles of *Legionella pneumophila* in bioaerosols after showering with (**b**) one representative STA showerhead and (**c**) one ECO showerhead (VC: GFP+/PI−; VBNC: GFP+/PI+; DC: GFP−/PI+); Overlays of the number of *Legionella pneumophila* (*y*-axis) in the reservoir (in blue), in the bag (in purple) and in the aerosols collected with the Coriolis (in grey) as a function of GFP fluorescence (*x*-axis) for (**d**) one representative STA and (**e**) one ECO showerhead analysis. GFP: Green Fluorescent Protein; PI: Propidium Iodide. The black lines indicate the thresholds for the fluorescence signal considered positive.

**Figure 3 ijerph-19-03285-f003:**
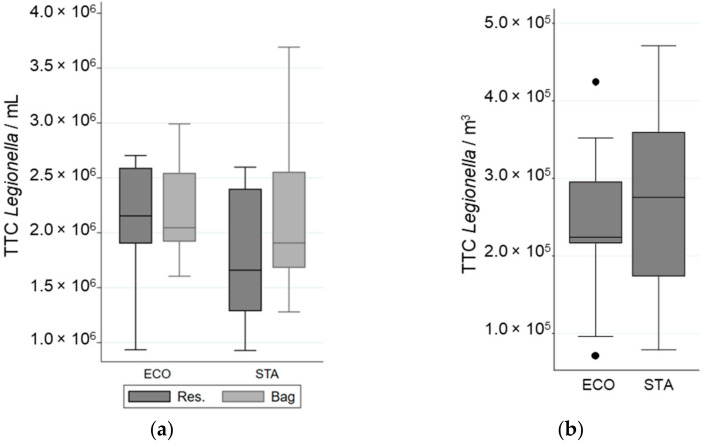
Variation in Total Cell Counted (TCC) of *L. pneumophila* in different samples when a “standard” (STA) or an “economic” (ECO) showerhead was used with an equal volume of water: (**a**) in the reservoir (Res.) and in the bag; (**b**) in the aerosols. The whiskers indicate the minimum and the maximum value, the box covers the values between the first and third quartile and the line in the box marks the median value.

**Figure 4 ijerph-19-03285-f004:**
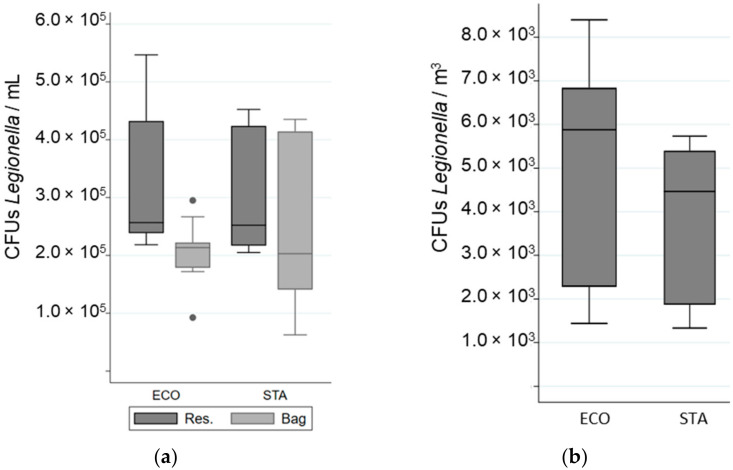
Variation in the number of Colony Forming Units (CFUs) of *Legionella* in different samples when a “standard” (STA) or an “economic” (ECO) showerhead was used with an equal volume of water: (**a**) in the reservoir (Res.) and in the bag; (**b**) in the aerosols. The whiskers indicate the minimum and the maximum value, the box covers the values between the first and third quartile and the line in the box marks the median value.

**Table 1 ijerph-19-03285-t001:** Characteristics of the showerheads used in the study.

Characteristic	Continuous Flow Showerhead (STA)	Water-Atomizing Showerhead (ECO)
Number of nozzles	51	6
Diameter of nozzle (mm)	0.8	1.1
Flow rate (L·min^−1^)	10.2	5.5
Spray angle (°)	5	36
Water pressure (bars)	1.2	2.4
Duration of the shower (s)	15	30

**Table 2 ijerph-19-03285-t002:** Mean number and proportion of viable and cultivable populations of *L. pneumophila* for all combined samples obtained by culture and by flow cytometric assay (FCA).

*Legionella* Physiological State	Data Considered	STA (*n* = 9)		ECO (*n* = 9)		*p* Value
		Mean	SD ^1^	Mean	SD ^1^	
Cultivable bacteria (VC) observed by culture	in the reservoir					
	CFU·mL^−1^	3.0 × 10^5^	1.1 × 10^5^	3.2 × 10^5^	1.3 × 10^5^	0.45
	Proportion of VC in TTC	13%	8%	14%	10%	0.17
	in the bag					
	CFU·mL^−1^	2.5 × 10^5^	1.4 × 10^5^	2.0 × 10^5^	5.8 × 10^4^	0.89
	Proportion of VC in TTC	11%	4%	10%	3%	0.51
	in the aerosols					
	CFU·m^−3^	3.8 × 10^3^	1.8 × 10^3^	4.9 × 10^3^	2.6 × 10^3^	0.16
	Proportion of VC in TTC	2%	1%	3%	3%	0.70
Viable bacteria (VC + VBNC) observed by FCA	in the reservoir					
	Cells·mL^−1^	1.7 × 10^6^	6.2 × 10^5^	2.2 × 10^6^	6.9 × 10^5^	0.12
	Proportion of viable bacteria in TTC ^2^	86%	3%	91%	3%	
	in the bag					
	Cells·mL^−1^	2.1 × 10^6^	7.6 × 10^5^	2.2 × 10^6^	5.1 × 10^5^	0.69
	Proportion of viable bacteria in TTC	91%	4%	91%	4%	0.63
	in the aerosols					
	Cells·m^−3^	2.0 × 10^5^	9.6 × 10^4^	1.8 × 10^5^	8.1 × 10^4^	0.84
	Proportion of viable bacteria in TTC	73%	3%	74%	3%	0.50

^1^ standard deviation, ^2^ TTC = VC + VBNC + dead bacteria.

## Data Availability

Not applicable.

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
