# Peer review of "Risk Exposure to Legionella pneumophila during Showering: The Difference between a Classical and a Water Saving Shower System"

_ijerph, 2022, doi:10.3390/ijerph19063285_

Round 1

Reviewer 1 Report

ijerph-1568847 Review Report

The manuscript presents an experimental study of the impact of the new water efficient atomization-induced showering technology on the aerosolization of Legionella as well their viability from the point of view of the human health risk. I believe it’s relevant research for ijerph audience due to a large public health concern and for the confidence of public in using technologies that are focused on water saving. The manuscript has following positives and negative aspects to it:

Positives:

  • Introduction and background is well researched and well written.
  • Research goals clearly stated, and methods used are clear and well written.

Drawbacks:

I have few comments and suggestions for improving this manuscript embedded throughout the file attached. Few major concerns are:

  • What is the -pressure in the glove box used for bathroom simulation and if that would impact the aerosolization of droplet as well as Legionella with it? It needs more explanation around it.
  • Results of statistical analysis (p-values etc.) are not presented.

Some of the results and discussion around it are overstated. For example, simulated artificial water reservoir with uniform and high concentration of Legionella is presented as strength while I think it’s a limitation. I have highlighted some of these concerns in the file attached. 

Author Response

The manuscript presents an experimental study of the impact of the new water efficient atomization-induced showering technology on the aerosolization of Legionella as well their viability from the point of view of the human health risk. I believe it’s relevant research for ijerph audience due to a large public health concern and for the confidence of public in using technologies that are focused on water saving. The manuscript has following positives and negative aspects to it:

Positives:

  • Introduction and background is well researched and well written.
  • Research goals clearly stated, and methods used are clear and well written.

Answer:  We thank the reviewer for these positives remarks

Drawbacks:

I have few comments and suggestions for improving this manuscript embedded throughout the file attached.

Answer:  We thank the reviewer for the comments formulated in the file. The answers are detailed here after.

Reviewer’s comment lines 25-27: How about a comparative result between two technologies?

Answer: the abstract was modified and information on the comparative result between the two technologies added

Reviewer’s comment Lines52-53: Is that a requirement or unintended consequence of low flow devices etc? Particularly longer water residence time.

Answer: We agree that this is an unintended consequence. The sentence was modified in consequence.

Reviewer’s comment Lines 60-61: Provide reference to this such as Water sense guidance from USEPA.

Answer: References were added

Reviewer’s comment Line 82: this is odd term. You want to say exposure risk.

Answer: The text was modified as suggested

Reviewer’s comment Line 97: What experience? Do you need this? I think unnecessary.

Answer: The sentence was modified as suggested.

Reviewer’s comment Table 1: Aren't these selected shower events too short? ON what basis these were selected?

Answer: The shower duration was limited by the volume of water that can be stored in the glove box (the reservoir capacity and the capacity of the bag to receive the water from the shower) as well as the water flow characteristics of the showerheads tested. These characteristics are described lines 125-138.

Reviewer’s comment Line 121: how about the pressure inside the glove box? Was it comparable to atmospheric pressure during the actual showering event? This will impact the aerosol formation quite a bit.

Answer: The pressure inside the glove box was comparable to the atmospheric pressure with a variation that did not exceed 0.5 kPa. The information was added in the material and methods lines 127-128.

Reviewer’s comment Line 157 : How this concentration of working solution was chosen?

Answer: The concentration of the working solution was chosen based on the quantification limit of flow cytometry in aerosols, i.e. greater than 1000 bacteria per mL.

Wouldn't this be little too much to simulate showering water scenario?

Answer: We agree that the showering water is generally contaminated by 1.1x103 CFU/L. However, for such a concentration of bacteria in water, the detection of viable bacteria in aerosols by flow cytometry is not reliable. Therefore, to be robust in our comparison of the two showerheads, we chosen a concentration of the working solution high enough to emit more than 1000 bacteria per mL of  aerosols sample.

Reviewer’s comment Figure 2: Apply label b and c properly. Either place them in the middle of the two graphs they are representing, change the arrangements how you place different graphs, or label them separately b, c, d, and e.

Answer: The modifications were done on figure 2

Formula (2): Which liquid we are talking about here? is it different than water?

Also I do not the basis of this formula. Am I missing something?

Answer: The meaning of the formula (2) was detailed in lines 232-235. Indeed aerosols were samples in a liquid.

Line 222: How many data point you have? DO you want to mention that here.

Answer: The experiment was run in triplicate with three distinct showerheads STA or ECO. The information was added in the Description of the experimental setup lines 147-148.

Line 265: Where are your results from statistical analysis? P-values etc for different comparison? Do you want to tabulate them in a table or include in table 2?

Answer: The p-values were added in the Table 2

Line 305-307: This is not necessarily an strength of the study but and approximate simulation of natural real world conditions. I'd say its a limitation, and definitely not a strength.

Answer: We avoid the usage of the word “strength” in the actual version

Line 309: I'd again say, this is limitation not a strength. The strength I'd say is something that would get you closure to real situation and not a hypothetical scenario.

Answer: We avoid the usage of the word “strength” in the actual version

Line 312: I'd say infection risk.

Answer: The modification was done

 Few major concerns are:

  • What is the -pressure in the glove box used for bathroom simulation and if that would impact the aerosolization of droplet as well as Legionella with it? It needs more explanation around it.

Answer: The pressure inside the glove box was comparable to the atmospheric pressure with a variation that did not exceed 0.5 kPa. The information was added in the material and methods. The glove box was used only as an airtight enclosure to protect the operators in front of the glove box from aerosolised Legionella.

  • Results of statistical analysis (p-values etc.) are not presented.

Answer: The p-values were added in the Table 2

Some of the results and discussion around it are overstated. For example, simulated artificial water reservoir with uniform and high concentration of Legionella is presented as strength while I think it’s a limitation. I have highlighted some of these concerns in the file attached. 

Answer: The modifications were made as suggested to avoid overstate the results

Reviewer 2 Report

In  the current manuscript the authors tried to experimentally determine the number of viable or cultivable L. pneumophila emitted during showering with one continuous flow and one water atomization showerhead. The latter method is a newly developed one which is thought to be eco-friendly. The secondary aim was to estimate the impact of the water atomizing  technology on Legionella survival. The authors developed experimental  systems to perform a the necessary risk assessment. The authors did not come up with any statistically significant difference between the two methods. The manuscript is generally well written and the language used seems adequate although certain improvements could be applied. The methodology used seems appropriate and the results are well discussed. Attached you will find a pdf file with a few comments on the document.

Author Response

In  the current manuscript the authors tried to experimentally determine the number of viable or cultivable L. pneumophila emitted during showering with one continuous flow and one water atomization showerhead. The latter method is a newly developed one which is thought to be eco-friendly. The secondary aim was to estimate the impact of the water atomizing  technology on Legionella survival. The authors developed experimental  systems to perform a the necessary risk assessment. The authors did not come up with any statistically significant difference between the two methods. The manuscript is generally well written and the language used seems adequate although certain improvements could be applied. The methodology used seems appropriate and the results are well discussed. Attached you will find a pdf file with a few comments on the document.

We thank the reviewer for the text editing and the comments emitted. All the typo were corrected as indicated lines 295-297 and Line 316 by the reviewer. The answers to the reviewer’s comments were detailed here after.

Reviewer’s comment Line 44.: Avoid beginning a phrase with a short cut.

Answer: We replaced the short cut at the beginning of the sentence as suggested.

Reviewer’s comment Line 111. Why did the authors use this antibiotic? 

Answer: The antibiotic usage is justify in the paragraph 2.3. The information was deleted in the paragraph 2.1 to avoid confusion.

Reviewer’s comment Line 153: Why +/- 2 and not +/-1?

Answer: The+/- 2°C is the maximum variation tolerated for an oven according to point 5.1 of standard ISO 8199:2018 - October 2018: 

5.1 - Uniformity of temperatures

The following accepted ranges of temperatures and their ranges for incubation or storage are applied, where appropriate for the intended target organism and unless otherwise required in the specific standard.

Incubation temperatures:           (22 ± 2) °C; (36 ± 2) °C; (44 ± 0,5) °C

The upper incubation temperature limits shall be followed strictly to ensure optimal growth. The lower temperature limits may be exceeded for short periods, e.g. due to opening the door of an incubator, but recovery to the operating temperature should be rapid.

Reviewer 3 Report

The manuscript deals with an interesting topic, investigation of viability of bacteria during showering and comparing two types of shower systems. However, the presentation has several weaknesses, the manuscript needs major revision.

Comments:

  1. p.1 l.37: "sever" --> "severe"
  2. p.3 Table 1, first row: "Number of nozzle" --> "Number of nozzles"
  3. l.122: "L.h-1" --> "L.h-1" ("-1" in superscript)
  4. p.4 l.147-148, 152, 166: acronym should be in brackets and full expression without brackets; full expression should precede acronym in brackets
  5. l.157: For claritry, numbers with errors should be in brackets, e.g. "(4.5±1.0)×104"
  6. p.5 l.183: "Scan 4000" --> "Scan 4000 instrument"
  7. l.187-188,190: acronym "PI" should be defined at the first occurrence of "propidium iodide"; "Invitrogen P3566" should be without brackets
  8. Fig. 2, l.197-202 (caption): The presentation of all five panels of Fig.2 is really poor, not clear for the reader. It should be drastically improved, taking into account the following requirements:
    (a) all five panels should bear a letter, i.e (a) to (e)
    (b) the panels look like outputs of a computer program and not scientific level figures. Physical quantities, units of measure and scales should be clearly given at abscissa and ordinate axes. Legends should not be filenames, but representative for the samples/circumstances
    (c) acronyms "SSC" and "FSC" are used without definition
    (d) description of axes should be moved from the figure caption to the figure itself, e.g. "number of Legionella pneumophila (y-axis)", "GFP fluorescence (x-axis)"
  9. p.6 l.207-217: The description of the conversions of liquid concentrations to air concentrations is really strange and needs to be clarified. Eqns (1) and (2) are not equations (nothing on the left-hand side); simple symbols should be used in equations not full text.
  10. l.294: "percentage" --> "proportion" (to be consistent with Table 2)
  11. l.234: See Comment 5
  12. p.7 Fig.4(b) and 5: numbers at y axis should be presented the same way as in Fig.4(a) - only one digit shown after decimal dot
  13. p.9 l.305-306: "homogenous" --> "homogeneous"
  14. l.318: "liters of water" --> "volume of water"
  15. l.320: "number of liters" --> "water volume"       

Author Response

The manuscript deals with an interesting topic, investigation of viability of bacteria during showering and comparing two types of shower systems. However, the presentation has several weaknesses, the manuscript needs major revision.

We thank the reviewer for the comments formulated. They were all addressed and the answer for each of them was detailed here after.

Comments:

  1. p.1 l.37: "sever" --> "severe"

Done

  1. p.3 Table 1, first row: "Number of nozzle" --> "Number of nozzles"

Done

  1. l.122: "L.h-1" --> "L.h-1" ("-1" in superscript)

Done

  1. p.4 l.147-148, 152, 166: acronym should be in brackets and full expression without brackets; full expression should precede acronym in brackets

      The full expression precede in the actual version the acronym

  1. l.157: For claritry, numbers with errors should be in brackets, e.g. "(4.5±1.0)×104"

Done

  1. p.5 l.183: "Scan 4000" --> "Scan 4000 instrument"

Done

  1. l.187-188,190: acronym "PI" should be defined at the first occurrence of "propidium iodide"; "Invitrogen P3566" should be without brackets

Done

  1. Fig. 2, l.197-202 (caption): The presentation of all five panels of Fig.2 is really poor, not clear for the reader. It should be drastically improved, taking into account the following requirements:
    (a) all five panels should bear a letter, i.e (a) to (e)
    (b) the panels look like outputs of a computer program and not scientific level figures. Physical quantities, units of measure and scales should be clearly given at abscissa and ordinate axes. Legends should not be filenames, but representative for the samples/circumstances
    (c) acronyms "SSC" and "FSC" are used without definition
    (d) description of axes should be moved from the figure caption to the figure itself, e.g. "number of Legionella pneumophila (y-axis)", "GFP fluorescence (x-axis)"

The modifications suggested were done and the figure replaced.

  1. p.6 l.207-217: The description of the conversions of liquid concentrations to air concentrations is really strange and needs to be clarified. Eqns (1) and (2) are not equations (nothing on the left-hand side); simple symbols should be used in equations not full text.

Sorry about that. The equations are indicated by symbols and their description was clarified

  1. l.294: "percentage" --> "proportion" (to be consistent with Table 2)

Done

  1. l.234: See Comment 5

Done

  1. p.7 Fig.4(b) and 5: numbers at y axis should be presented the same way as in Fig.4(a) - only one digit shown after decimal dot

The figures were changes as suggested

  1. p.9 l.305-306: "homogenous" --> "homogeneous"

Done

  1. l.318: "liters of water" --> "volume of water"

Done

  1. l.320: "number of liters" --> "water volume"  

Done